# Development of New PCR Assay with SYBR Green I for Detection of *Mycoplasma, Acholeplasma*, and *Ureaplasma* sp. in Cell Cultures

**DOI:** 10.3390/diagnostics11050876

**Published:** 2021-05-14

**Authors:** Jolanta Krzysztoń-Russjan, Jakub Chudziak, Małgorzata Bednarek, Elżbieta Lidia Anuszewska

**Affiliations:** 1Department of Biochemistry and Biopharmaceuticals, National Medicines Institute (NMI), 00-725 Warsaw, Poland; e.anuszewska@nil.gov.pl; 2Internship at the Department of Biochemistry and Biopharmaceuticals, NMI Chelmska 30/34 Str., 00-725 Warsaw, Poland; jmchudziak@gmail.com (J.C.); bednarekmj@gmail.com (M.B.)

**Keywords:** *Mycoplasma*, *Acholeplasma* and *Ureaplasma* sp. detection, qPCR validation, cell cultures, quality control

## Abstract

*Mycoplasma*, *Acholeplasma*, and *Ureaplasma* sp. are atypical bacteria responsible for in vitro cell culture contaminations that can warp the results. These bacteria also cause human and animal infections and may lead to chronic diseases. In developed polymerase chain reaction (PCR) in this study a quantitative PCR with SYBR Green I fluorochrome was applied to facilitate the *Mycoplasma*, *Acholeplasma*, and *Ureaplasma* sp. DNA detection and identification. Screening Test-1 v.1 (triplex qPCR) allowed for the detection of 11 species. Test-1 v.2 (three single qPCRs) pre-identified three subgroups, allowing for the reduction of using single qPCRs in Test-2 for species identification. The range of both tests was consistent with pharmacopeial requirements for microbial quality control of mammal cells and included detection of *M. arginini*, *M. orale*, *M. hyorhinis*, *M. fermentans*, *M. genitalium*, *M. hominis*, *M. pneumoniae*, *M. salivarium*, *M. pirum*, *A. laidlawii*, and *U. urealyticum*. Limit of detection values varied between 125–300 and 50–100 number of copies per milliliter in Test-1 and Test-2, respectively. Test-1 and Test-2 showed fully concordant results, allowed for time-saving detection and/or identification of selected species from *Mycoplasma*, *Acholeplasma*, and *Ureaplasma* in tested cell cultures.

## 1. Introduction

Bacteria from *Mycoplasma*, *Acholeplasma,* and *Ureaplasma* genera are common contaminants of cell cultures, including the most frequently appearing: *M. arginini*, *M. hyorhinis*, *M. orale*, *M. salivarium*, *M. fermentans*, *M. hominis*, *A. laidlawii*, as well as others occasionally found in cell cultures: *U. urealyticum*, *M. pneumoniae*, and *M. pirum* [1,2,3]. Several possible causes of these contamination in cell cultures have been previously described, such as bacterial residues in primary cultures, culture media, fetal bovine serum, or no use of aseptic principles by laboratory staff. Cross-contaminations between infected cultures can happen during running multiple cell cultures simultaneously or using contaminated laboratory equipment such as pipettes, laminar flow cabinets, and CO_2_ incubators [4,5,6]. 

The majority of *Mycoplasma*, *Acholeplasma*, and *Ureaplasma* sp. are a part of normal human or animal bacterial microflora; however, some of these bacterial species sometimes are responsible for developing acute or chronic infections with an opportunistic character [7,8,9]. The *Mycoplasma* and *Ureaplasma* genera from the *Mycoplasmatales* order and the *Acholeplasma* genus from the *Acholeplasmatales* order represent the *Mollicutes* class [10]. These bacteria differ from the other bacterial microorganisms by the cell wall absence and the reduced genome size. The smallest genome size that was found in *M. genitalium* with 577–590 kilobase pairs (kb) [11,12] in contrast to *Escherichia coli* genomes ranging between 4.5 and 5.5 megabase pair (Mbp) [13]. The size of these bacteria cells (0.15–0.3 µm) do not allow to detect it by indirect microscopic observation. Therefore fluorescent DNA staining with 4′,6-diamidino-2-phenylindole (DAPI), or Hoechst dyes are usually used. DAPI staining enables a quick but unspecific detection using a fluorescent microscope with an excitation wavelength of approximately 355 ± 5 nm and an emission wavelength of 455 ± 5 nm [14]. *Mollicutes* contamination is observed then in a microscope as shining points localized in the cytoplasm or at the cell’s edge [7]. 

Apart from the methods mentioned above, several other types of methods were applied in detection of these bacteria, including microbial cultures using selective broth or agar media, and also nucleic acid hybridization with a probe specific for the evaluation of mycoplasma’s rRNA [4], immunofluorescence assays with particular monoclonal antibody dyes for the first step of *Mycoplasma*, *Ureaplasma*, and *Acholeplasma* species identification [15], enzyme-linked immunosorbent assay (ELISA) for the detection of *Mycoplasma* species-specific antigen [16] or PCR base techniques for detection of species or genus specific nucleic acid fragments [6,7,17]. 

Currently, PCR is the most popular and accessible molecular biology technique. Since 1985 [18], it has been developed to displace other techniques in diagnostics of different diseases. Now the use of PCR for detection and identification of *Mycoplasma*, *Acholeplasma*, and *Ureaplasma* sp. in cell cultures or in other clinical samples is considered as more valuable than the use of microbial growth cultures. It is related primarily with higher percentage of detection and discriminatory power [17,19], significantly shorter time of obtained results [8], and lower costs in contrast to microbial culture methods. A quantitative PCR-based method which employ primers detecting DNA sequences from *U. parvum* and *U. urealyticum* (*gap* gene) and from *M. genitalium* and *U. hominis* (*ureC* gene) has recently been reported in a study conducted on clinical samples, including tissue (chorionic villi) and blood samples from pregnant females [20]. Additionally a highly sensitive multiplexed qPCR configured with digital microfluidic platform for the detection of DNA *M. pneumoniae* and other microorganisms in discrete droplet format designated for liquid samples has also been reported [21]. Among PCR technique types applied for *Mycoplasma*, *Acholeplasma* and *Ureaplasma* detection in cell cultures, a real-time PCR was shown as superior with the sensitivity, specificity, accuracy, predictive value of positive and negative results [17]. 

Nowadays, loop-mediated isothermal amplification (LAMP) assay is a novel DNA amplification technique that uses 4–6 primers to recognize specific regions on the target DNA. It is possible to detect a few up to 10^9^ copies of target region DNA in less than an hour. Although the first it was described in 2000 by Notomi et al. [22] an application of this assay for detection of *Mycoplasma* sp. contamination in cell cultures was described by Soheily et al. a few years ago [23]. The developed LAMP assay detected some *Mycoplasma* species and differed from other PCR assays by isothermal conditions of reaction and possible other type of detection with naked eye inspection or in the Loopamp real-time turbidimeter. 

Development of reliable PCR methods is especially needed for microbial quality control of cell cultures as well as for contamination evaluation of other types of biological samples caused by difficult microorganisms. That methods can be also a part of infection diagnostics after additional validation.

Our study aimed to develop a sensitive and specific assay for routine detection and identification of *Mycoplasma*, *Acholeplasma*, and *Ureaplasma* selected species during a part of microbial quality control of mammalian cell cultures according to pharmacopeial requirements.

## 2. Materials and Methods

### 2.1. Cell Lines Cultivation and Positive Controls

We examined in this study eighty one samples of seventeenth different cell lines deposited in the National Medicines Institute for *Mycoplasma*, *Acholeplasma*, and *Ureaplasma* contamination using Test-1 and Test-2. Following cell lines were included, as: human epithelial cells from cancer lung (A549, *n* = 1), human epithelial cells (WISH, *n* = 3), human cervical carcinoma (KBV, *n* = 4), human osteosarcoma cells (MG63, *n* = 9), mouse fibroblasts (L929, *n* = 19), human epithelial adenocarcinoma cells (HeLa, *n* = 14), human melanoma cells (ME18, *n* = 2; MER, *n* = 2), human B lymphoid cells (GM13509, *n* = 7; GM1446, *n* = 6), human promyelocytic leukemia cells (HL-60, *n* = 4), normal human osteoblasts (hFOB 1.19, *n* = 3), breast cancer cell line (MCF-7, *n* = 1), human fibroblasts (GM21756, *n* = 1; BJ, *n* = 2), mouse T lymphocytes (TK-1, *n* = 2), and bovine kidney cells (MDBK, *n* = 1). ME18 and MER cells, that were acquired from the National Medicines Institute archival collection [24]. HeLa, L929, hFOB 1.19, HL-60, TK-1, WISH, A549, MDBK, and MCF-7 were purchased from the ATCC collection (USA), MG63 from the EACACC collection (UK), KBV1 from the DSMZ collection (Germany), GM03651, GM21756, GM1446; GM13509 from the Coriell Institute collection (USA). All adherent cells cultured in growth medium composed of Eagle’s Minimum Essential Medium (EMEM) supplemented with 10% FBS and 1% antibiotics solution (penicillin, streptomycin, amphotericin B). B-lymphoid cells cultured in RPMI medium, 15% fetal bovine serum (FBS), and 1% antibiotics solution. Cell culture media and supplements were purchased from Lonza (Walkersville, MD, USA). Growth conditions included 37 °C and a humidified atmosphere of air containing 5% CO_2_.

As positive controls, we used in this study genomic DNA (gDNA) standards of aimed species at concentration of 0.05 ng per reaction. All DNA templates we purchased from Minerva Biolabs GmbH (Germany) (Appendix A). The other DNA templates were derived from standard strains purchased from the National Collection of Type Cultures (NCTC) and the Culture Collection of Public Health England (Appendix A). The initial concentration of all gDNA standards was 0.1 ng/µL. All templates were suspended in 10 mM Tris buffer, pH 8.4 at a concentration of 0.1 ng/mL, and stored at −20 °C. A certificate of analysis confirmed the quantity of each DNA samples.

### 2.2. Primer Design, In Silico Specificity Determination 

We performed the first search for a specific DNA region occurring only in selected species according to this study’s aim using BioEdit software [25]. The analysis included 16S-23S Internal Transcribed Spacer (ITS) region, *16S rRNA*, *23S rRNA,* and other genes across mentioned above *Mycoplasma*, *Acholeplasma*, and *Ureaplasma* species to select DNA fragment typical for all selected species of interest (Appendix A). Three different specific DNA fragments were chosen, and three primer pairs, including M1, M2, M3 forward (Fw), and reverse (Rv) were designed for genetic in silico identification using Beacon Designer software (BDS) (Premier Biosoft Int., v. 7.91), according to the general primer requirements [26,27]. For amplicon sequences analysis in silico, the primary local alignment search tool (BLAST) from the browser (http://www.ncbi.nlm.nih.gov/BLAST/ (accessed on 12 May 2021) was used. We compared the target sequences with those found in the GenBank sequence database provided by NCBI matching to all available bacterial genomes, humans, bovine, mice, and rats. 

Validation of Test-1 and Test-2 included applicability, specificity, sensitivity, precision, accuracy, and robustness, according to available requirements [26,27,28,29,30]. Finally, microbial growth culture tests were performed only with DNA derived from MG63 cells to verify DAPI and qPCR results. 

### 2.3. Primer Concentrations and Thermal Profile Optimization

Several combinations of forward—Fw (5′→3) and reverse—Rv (3′→5′) primer concentrations including 100, 150, 200, 300 nM were used with 10.0 μL KAPA SYBR FAST qPCR Master Mix (2×), ROX Low option (Kapa Biosystems, Boston, MA, USA), and 1.0 μL of template gDNA input containing 0.05 ng per reaction. Negative control (PCR-grade water) was added to each combination of primer concentrations. The final volume of a single qPCR reaction was achieved by adding PCR water up to a final volume of 20.0 μL.

The study was performed using the MxPro 3005P QPCR System (Agilent Technologies, Santa Clara, CA, USA). Thermal profile optimization for three primer pairs selected for Test-1 with sequences presented in Appendix A was performed in different experimental variants that varied in annealing temperature selected from the range of 60–65 °C. In Test-1, optimal M1, M2, and M3 primer pairs concentrations were combined with 0.05 ng (*M. orale*, *M. hyorhinis*, and *M. genitalium*) DNA standards per single reaction, respectively (Appendix A). In Test-2, all primer pairs presented in Appendix A at the optimal concentration were combined with appropriate 0.05 ng DNA templates for thermal profile optimization. 

### 2.4. qPCR Efficiency and Linearity Evaluation

The efficiency of the qPCRs used in Test-1 and Test-2 was evaluated based on DNA standard curves obtained for serial dilutions of DNA standards (Appendix A). Each *Mycoplasma/Acholeplasma/Ureaplasma* sp. DNA standards were diluted in five steps by ten-fold serial template dilutions to give the ranges presented in Table 1 and Table 2. DNA standards expressed in ng were next converted into number of copies (c) by MxPro software. The molecular weight required for unit conversion was obtained using Polynucleotide Molecular Weight Calculator presented on the website http://scienceprimer.com/nucleotide-molecular-weight-calculator (accessed on 12 May 2021). *M. orale* DNA standard concentration was only initially expressed in cn/mL. It was additionally diluted in five steps by ten-fold serial template dilution to give the ranges between 2.50 × 10^5^ ÷ 2.5 × 10^0^ c in Test-1 (Table 1) and 5.0 × 10^5^ ÷ 5.0 × 10^0^ c in Test-2 (Table 2). The efficiency (E) was calculated based on the formula generated by the MxPro 3005P software after plotting standard curves for each tested primer pair (MxPro software v.4.10, Agilent Technologies, Santa Clara, CA, USA).

The E values of qPCR were evaluated individually for gDNA standards of 10 selected species belonged to *Mollicutes* and additionally for the *M. orale* c standard. Each dilution was performed in triplicate. The experiments were repeated at least three times for each DNA standard. During PCR efficiency evaluation, two parameters were simultaneously determined, including R square (RSq or R^2^) and slope values, reflecting linearity and inclination of the standard curve. The characteristic details of the standard curves were described by a curve equation that expressed the relationship between the *C*_t_ value and the appropriate amount of DNA from the dilution series.

### 2.5. Test-1 and Test-2 Analytical Specificity Evaluation

Analytical specificity for Test-1 was performed for M1, M2, and M3 primer pairs with their optimal concentrations presented in Table 1 using optimal thermal profile (shown above) and 0.05 ng each of analyzed three gDNA. The comparison between triplex qPCR with all primers M1, M2, and M3 Fw and Rv in one tube (Test-1 v.1) and three single qPCRs (Test-1 v.2) with one of each pair of primers respectively were made to specificity analysis of received qPCR products. For comparison, specificity between Test-1 v.1 and -v.2 *M. hominis* and M1, *M. fermentans* and M2, *M. pneumoniae* and M3 were used. T (Appendix A) evaluate of Test-2 analytical specificity, ten specific qPCR reactions were performed in duplicate with gDNA 0.05 ng templates of *M. arginini*, *M. orale*, *M. hyorhinis*, *M. fermentans*, *M. genitalium*, *M. hominis, M. pneumoniae*, *M. salivarium*, *A. laidlawii*, and *U. urealyticum* with each pair of primers, containing sequences presented in Appendix A with concentrations shown in Table 2. Both Test-1 and -2 specificity examinations were repeated twice. The specificity of received qPCR products was analyzed using melting curve results. Specific pick with a maximum of melting temperature (Tm) received for individual pair of primers confirmed each PCR’s specificity.

To verify M1, M2, and M3 primer pairs cross-reactivity, the next experiment was performed using 1.0 ng gDNA per reaction from a phylogenetically closely related bacterial species. The analysis included *Enterococcus faecalis*, *Lactobacillus fermenti*, and then non-related bacterial species such as *Salmonella enteritidis*, *Candida albicans*, *C. tropicalis*, *Saccharomyces cerevisiae*, and also with 10.0 ng DNA per reactions of mammal species: *Mus musculus* (ATCC) and *Homo sapiens* (Coriell Institute for Medical Research, Camden, NJ, USA). 

### 2.6. Analytical Sensitivity—Limit of Detection (LOD) and Limit of Quantification (LOQ)

We defined the LOD values for Test-1 and Test-2 based on the standard curves performed with DNA of representative species from tested *Mycoplasma, Acholeplasma*, and *Ureaplasma* sp. LOD values expressed the lowest detectable cn value obtained for 95% of all (true positive) replicates test positive with acceptable accuracy and precision [30]. 

We defined the LOQ as the lowest number of copies of isolated DNA, which allowed for the reliable quantification of a specific amplicon, with acceptable accuracy and precision for 95% of all (true positive) replicates test positive sufficient accuracy and precision. [30]. 

### 2.7. Test-1 Precision and Robustness Determination

Measurements for precision determination made across at least three independent run in triplicate. Precision was determined by relative standard deviation (RSD) calculation using the formula RSD = S/X; S—standard deviation; X—mean value and calculated for within-run (RSDw) and between-run precision (RSDb) determination, based on the *C*_t_ value differences obtained for all specific amplicons in Test-1 (Appendix A). 

DNA extracted from the B-lymphoid cell line GM1447 used as a template was diluted 2-fold in a five-step serial dilution to give a final concentration range of 1.2 × 10^0^ ÷ 7.5 × 10^−^^2^ ng/mL (Appendix A). Next each dilution was spiked with *M. orale* DNA serially diluted to a range between 2.5 × 10^5^ ÷ 2.5 × 10^0^ c/mL and combined with M1, and between 2.25 × 10^5^ ÷ 2.25 × 10^0^ c/mL with M2 primers at the optimal concentration. In the case of M3 primers, master mix samples were additionally spiked with *U. urealyticum* DNA serially diluted to a range between 5.0 × 10^4^ ÷ 5.0 × 10^0^ c/mL. In this analysis qPCR efficiency and specificity for M1 M2 and M3 were evaluated (Appendix A). 

The robustness (R) of Test-1 we evaluated by assessing the influence of plastic tube types on the level of detection of a known amount of DNA, expressed by the results of a precision evaluation. The effect of plastic tube type on assay sensitivity was defined as an average *C*_t_ value and calculated for qPCRs performed in plastic tubes obtained from each of the three qPCR tube manufacturers (Appendix A). 

### 2.8. Diagnostic Sensitivity, Specificity and Accuracy Evaluation Using PromoKine qPCR Test Kit I/RT, Variant C

Thirty DNA samples isolated from different cell lines were tested by commercial PromoKine qPCR Test Kit I/RT Variant C according to the manual manufacturers (PK-CA91-3025C). 500 μL of cell culture supernatant was transferred into sterile DNA-free reaction tube and incubated at 95 °C for 5 min, then centrifugated at 10,000× *g* for 25 s. Finally 2 μL of the extract was used as a template for qPCR that was performed using manual instructions and MxPro 3005P QPCR System with appropriate settings. 

To obtain diagnostic sensitivity and specificity 30 DNA samples derived from selected cell cultures (Table 3) were additionally analyzed by both Test-1, Test-2 described in this study and commercial PromoKine qPCR Test Kit I/RT, Variant C. We compared qualitative results obtained by both tests PromoKine qPCR Test Kit I/RT and Test-1 for the diagnostic sensitivity and specificity evaluation according to Kralik et al. [30].

The tests’ accuracy showed the level agreement of qualitative results obtained in both methods for the same tested cell samples. All obtained results achieved for M1, M2, and M3 tested primer pairs of Test-1 and obtained in Test-2 were compared with the results obtained with commercial PromoKine qPCR Test Kit I/RT, Variant C that was used in this study as a referenced method. Defining accuracy, we considered four primary categories applicable in binary classification tests (recommended for qualitative detection). We analyzed thirty samples by Test-1 and Test-2, calculating their concordance with the reference method according to the formula recommended by Kralik et al. [30]. 

### 2.9. DAPI Test and Mycoplasma Culture on Agar

Human osteosarcoma cells (MG63) and mouse fibroblasts cells (L929) were detached using a 0.5 g/L trypsin 1:250 and 0.2 g/L Versen (EDTA) solution (Lonza, Walkersville, MD, USA) and immediately separated into single cells by pipetting. Next, the growth medium with FBS added (1:1 ratio) to stop trypsin activity. The received cellular suspension was centrifuged for 3 min at 500 RPM. The cell sediment rinsed with Dulbecco’s Phosphate-Buffered Saline (DPBS) with calcium and magnesium, and after centrifugation, the pellet was suspended in a working solution of 1 μg/mL DAPI (AppliChem Gmbh, Darmstadt, Germany Germany) in methanol and incubated at 37 °C for 10 min. Cells were rinsed with DPBS and examined under an inverted Nikon microscope (Nikon Instruments Europe BV, Amstelveen, The Netherlands) using a DAPI filter (emission at: 435–485 nm and excitation: 340–380 nm wavelength) at 400× magnification.

We have mentioned above, MG63 and L929 72 h old cells with the same cell number concentration and cell dilution ranges, staring from 1 × 10^4^ number of cells/mL (1×) to inoculate Mycoplasma Agar Base supplemented with mycoplasma selective supplement-G (MABG) (Oxoid Ltd., Hampshire, England) to test mycoplasma growth. Then, 1 mL of undiluted (1×), 10^−^^1^ (10×) and 10^−^^2^ (100×) diluted cell suspension were added into MABG plates in two independent repeats. Agar plates incubated at 37 °C in 5% CO_2_ and evaluated after 1 and 2 weeks. MABG was prepared according to the manufacturer’s instructions. Evaluation of bacterial growth was performed after one and two weeks periods under direct detection using manual magnifier (~20× magnification) and inverted microscope using 400× microscope (Olympus, Hamburg, Germany) magnification.

## 3. Results

### 3.1. The Applicability of Test-1 and Test-2

The first—screening test, Test-1, was designed for the simultaneous detection of 11 selected *Mycoplasma/Ureaplasma/Acholeplasma* species using three primer pairs, including M1, M2, M3 Fw and Rv primers in specific triplex qPCR reaction (Test-1 v.1) (Scheme 1a) or three single qPCRs (Test-1 v.2) (Scheme 1b). Test-2 was arranged for optional species identification in case of positive qPCR results obtained in Test-1. Test-2 was developed as an option for species identification of previously selected *Mycoplasma/Ureaplasma/Acholeplasma* species (Scheme 1c). Both tests are applicable for the same 11 *Mycoplasma/Ureaplasma/Acholeplasma* species in specific qPCRs for *M. arginini*, *M. orale*, *M. hyorhinis*, *M. fermentans*, *M. genitalium*, *M. hominis*, *M. pneumoniae*, *M. salivarium*, *M. pirum*, and two closely related species, *A. laidlawii,* and *U. urealyticum.*


#### 3.1.1. Primers Design 

Based on the DNA sequence analysis performed in silico using BioEdit software, three different fragments of the *16S rRNA* gene were distinguished (Figure 1a–c, Table 1). Using Beacon Designer v. 7.91 software (Premier Biosoft International, San Francisco, CA, USA), three pair of primers, M1, M2, M3 designed for the identification of all species targeted in this study. They formed Test-1 for genus level detection of all *Mycoplasma*, *Acholeplasma*, and *Ureaplasma* species chosen for this study. M1 primer pairs presented 100% alignment, both forward (Fw) and reverse (Rv), with the *16S rRNA* gene sequences of *A. laidlawii*, *M. arginini*, *M. hominis*, *M. salivarium* and *M. orale* (Figure 1a). M2 displayed the same as M1 properties for *M. arginini*, *M. hominis*, *M. orale*, *M. salivarium*, except *A. laidlawii* (Figure 1b) an additionally for *M. hyorhinis*, *M. fermentans*, and M3 for *M. pneumoniae*, *M. pirum*, *M. genitalium*, and *U. urealyticum* (Figure 1c). 

Tm values designated for Test-1 primer pairs ranged between 59.7–61.6 °C (Fw/Rv) for M2 and 61.3–62.0 °C (Fw/Rv) for M1 Fv/Rv primers (Appendix A). They were slightly higher than recommended by the primer guidelines for obtaining the best results of 52–58 °C (Premier Biosoft International, San Francisco, CA, USA). It is generally accepted that primers with melting temperatures above 65 °C have a tendency for secondary annealing. The ΔG values calculated for hairpins and dimers for M1, M2 M3, and ACTB primers presented in this study were significantly lower than mentioned in guidelines and are listed at Appendix A. It was favorable to obtain one specific PCR product with a characteristic Tm value, depending on the specific DNA template (Figure 1; Table 1). 

Ten primer pairs were designated for the species identification mentioned as the aim of the study. To increase the specificity of designed primers, quality analysis of genes had been previously performed to find unique species-specific genes through sequence analysis. For species identification in Test-2 individual genes were selected as follows: *A. laidlawii*, family RNA methyltransferase (*trmA*) gene; *M. hyorhinis*, ethanolamine(*eutD*) gene; *M. fermentans,* (*dnaA*) gene; *M. hominis*, ACP phosphodiesterase (*acpD/ACP*) gene; *M. orale*, *M. salivarium*, *M. pirum*, DNA polymerase b-subunit (*rpoB*) gene; *M. genitalium*, tubulin-like protein (*ftsZ*) gene; *M. genitalium*, glyceraldehyde-3-phosphate dehydrogenase *(gap*) gene and for *U. urealyticum* DNA cytosine methylase (*dcm*) gene. The details related to species-specific primers used in Test-2 are presented in Table 2 and include selected data, such as a specific % of starter rate, calculated primer melting temperature, %GC content, ΔG values calculated for hairpins and dimers. Tm values calculated for primers used in the Test-2 were lower than those for Test-1 and varied in pairs between 55.3–55.5 °C for *A. laidlawii* and 58.5–58.6 °C for *M. orale*. Tm values corresponded to lower DG values for hairpins and dimers, found to be below 1.9, except for rpoB Rv primer hairpins with ΔG = 2.1 in the case of *M. salivarium* and also with lower PCR product size obtained in Test-2 (Appendix A).

#### 3.1.2. Primer Concentration, Thermal Profile Optimization and Efficiency

The optimization results for all Fw/Rv primer pairs designed in Test-1 and Test-2 are presented in Table 1 and Table 2, respectively, Figure 2 and Figure 3, and Appendix A. There were included optimal primer concentrations and also specificity, efficiency and linearity of each qPCR. The dynamic range values were shown in ng and in cn values wherein the cn values were obtained after conversion of ng values, considering the amplicon size and sequence, according to the formula described by http://scienceprimer.com/nucleotide-molecular-weight-calculator (accessed on 12 May 2021) and MxPro software. The Rsq values obtained for M1-M3 primers ranged between 0.980 and 0.983, meet the linearity criteria of ≥0.98 and received a confidence level of at least 95% (Figure 3; Table 1) [29]. The lowest R^2^ coefficient values were found for M1 primers in Test-1 and coincided with the lower dynamic range obtained for *A. laidlawii* and *M. orale* DNA templates as well as with the highest level of sequence variability found for these microorganisms, expressed in the number of different nucleotides. Based on the slopes and the linearity values describing the standard curves, the amplification efficiencies were found to range from 90.3 to 98.3% for all used primers in Test-1 (Figure 3) (derived from the formula E = 10^(-1/slope)^ − 1. Similarly Test-2 results present Table 2 with Rsq values ranged between 0.986 and 0.998 and E values ranged between 90.9 and 101.9%. All obtained results also met the appropriate criteria. 

The optimal thermal profile found in the course of this study included one cycle with initial polymerase activation at 95 °C for 2 min., and 40 cycles with annealing at 65 °C (Test-1) or 60 °C (Test-2) for 25 s, extension at 72 °C for 1 s and a single end cycle for melting temperature determination with 1 min at 95 °C, 30 s at 55 °C and 30 s at 95 °C. The two thermal profiles differed only in annealing temperature values, (65 °C in Test-1; 60 °C in Test-2). The increase of the annealing temperature to 65 °C in Test-1 was necessary to elevate the specificity of the qPCR results by ensuring the formation of only one peak on dissociation curve and was related to the elevation of the Test-1 sensitivity after increasing the primer concentration. 

#### 3.1.3. Test-1 and Test-2 Specificity and Analytical Sensitivity

The optimal PCR conditions made it possible to obtain a specific amplification product. The specificity of amplicons was manifested by maximum of Tm value that was expressed as a single peak for the majority of DNA standards or by two peaks (with two Tm values) in the case of *A. laidlawii* and M1 primers (Appendix A, Table 1) shown in dissociation curve. Tm values characteristic for all PCR products were shown in Table 1, Figure 2 and Appendix A for Test-1. No qPCR products were found with all tested primers and nonrelated DNA standards (*S. enteritidis*, *C. albicans*, *C. tropicalis*, *S. carlsbergensis*, *H. sapiens*, and *M. musculus*). There was no differences (in Tm values) between specificity of Test-1 v.1 and v.2, however slight differences was observed in *C*_t_ values obtained for *M. hominis* (M1) and *M. pneumoniae* (M3) (Figure 2). The absence of amplicons and with non-related DNA standards and with negative controls confirmed analytical specificity of M1, M2, and M3 primers. Table 2 and Appendix A present specificity results for Test-2. Tm values for achieved amplicons slightly varied depending of the DNA standard concentration that is reflected by SD values below ±0.5 °C. Negative controls (NC) were performed without template DNA but with each of the tested primer pairs designated for Test-1 and Test-2 in separate reactions and resulted in no amplification of specific products. 

The analytical sensitivity of both tests designed in this study is expressed by LOD values determined for all tested primer pairs (Table 1 and Table 2). 

LOD values obtained for Test-1 amounted 125–300 c/mL (<10 c/PCR reaction) for M1 and 250 c/mL (≤5 c/PCR reaction) for M2, and M3 primers. The LOD values obtained for Test-2 achieved 50–100 c/mL (<5 c/PCR) for *M. arginini, M. orale*, *M. fermentans*, *M. genitalium*, *M. pneumoniae*, *M. salivarium*, *M. hyorhinis*, *U. urealyticum*, *A. laidlawii*, and *M. hominis* (Table 2). LOD values obtained for Test-2 amounted <100 c/mL that (2 cn/PCR). 

The LOQ values obtained M1, for M2 and M3 primers are presented in Table 1, and there are comparable with LOD values but for M1 were a little bit higher. In Test-2 almost all species specific primers characterized slightly higher LOQ values (Table 2). 

#### 3.1.4. Precision and Accuracy 

The analytical precision obtained for Test-1 presents Appendix A. It was calculated for each independent experiment and expressed by coefficients of variation (%CV) for within-run precision (RSDw) and between-run precision (RSDb) for three experiments performed in different days with ACTB primers as Internal Control (IC) using a serial dilution of human DNA. The RSDw values for ACTB varied between 0.25 and 0.89, 0.41 and 1.08, and 0.7 and 2.49% in contrast to higher RSDb values that varied between 3.13 and 8.01. 

The precision for M1, M2, and M3 primers presents Appendix A. It was calculated similarly using for RSDw evaluation with *M. orale* DNA dilutions for M1 and M2, and *U. urealyticum* DNA dilutions for M3. RSDw (intra-assay) values for M1 ranged respectively between 1.28 and 1.98 for M2, 0.70 and 2.95, and for M3 0.65 and 2.12% (Appendix A). In contrast to RSDw (intra-assay) values below 5%, RSDw (inter-assay) values varied between 5.49 and 6.69 (Appendix A). The variation of all RSD values found for Test-1 was considerably below the required level of 25% [29,30]. The highest differences often concerned qPCR results with extremely lower template concentrations. 

To evaluate human (h) DNA concentration influence on *Mycoplasma*, *Acholeplasma*, and *Ureaplasma* detection in Test-1 efficiency and specificity of qPCR with M1, M2, and M3 were evaluated for selected *Mycoplasma* and *Ureaplasma* DNA standards. The results are presented in Appendix A. Efficiency for M1 primers obtained for *Mycoplasma* DNA standard and hDNA standard spiked with *Mycoplasma* amounted to 95.9 and 99.7%, respectively, for M2 103.1 and 104.6%, respectively. Efficiency obtained for M3 primers and *Ureaplasma* DNA standard and also hDNA standard spiked with *Ureaplasma* DNA amounted to 99.9 and 95.7%. Accuracy and precision agreed with general guidelines (between 70 and 120% for quantitative methods with recoveries) and fluctuated between 95 and 100%. It was shown no influence of hDNA for specificity of M1, M2, and M3 amplicons in 1.2 × 10^0^–7.5 × 10^−2^ ng/µL hDNA concentration range. For QC performed in our study 10 ng DNA templates derived from tested human cells were analyzed.

The results agreement between qPCR developed in this study and referenced method, the PromoKine qPCR Test Kit I/RT, Variant C, commercial test based on qPCR was obtained. Relative accuracy (RA) was calculated according to the formula (TP + TN)/(TP + TN + FP + FN) × 100% [29] and amounted 100% for Test-1 and Test-2 (Table 4). The comparison of results obtained in this study using Test-1 and Test-2 with referenced test, PromoKine qPCR Test Kit I/RT, Variant C concerned thirty different cell samples with 12 other cell lines showed in Table 3. The advantage of the presented method over the used commercial test but applied in this study as the “gold standard” is that developed assay allows at the first *Mycoplasma*, *Acholeplasma*, and *Ureaplasma* sp. DNA detection in the screening Test-1 and next, the species identification in Test-2 according the requirements of Ph. Eur for mammalian cell cultures. 

#### 3.1.5. Diagnostic Sensitivity, Specificity and Robustness

Sensitivity and specificity parameters were calculated according to the formulas described below using data from Appendix A. Sensitivity: true positives/(true positives + false negatives); specificity: true negatives/(true negatives + false positives); predictive value of a positive results: true positives/(true positives + false positives) and predictive value of a negative results: true negatives/(true negatives + false negatives) respectively. Sensitivity, specificity, and robustness calculated for Test-1 showed full concordance with PromoKine qPCR Test Kit I/RT, Variant C and all accuracy parameters amounted 100% [30]. 

This study’s robustness was determined to measure the variation of results obtained under different conditions, such as different plastic tube types for qPCR delivered by three other manufacturers (Appendix A). Therefore in this study, we evaluated the influence of plastic tube type on the *C*_t_ results obtained for Test-1. It was assessed through precision determination for M1, M2, M3, and ACTB primers and for different template standards performed in three types of plastic tubes. Precision was expressed in CV% values that corresponded to the precision measured for each individual tube type RDS1, 2, 3, and RSD1-3 values to reflect the precision obtained for all type tube types together.

The lowest CV% was <8.55% determined using the *C*_t_ results obtained for tube-type I that corresponded with the lowest *C*_t_ value for all of the primer pairs and templates compared to the results obtained for tube-type III with slightly higher CV % and the highest *C*_t_ values. The highest CV% values variation was observed after the final comparison obtained using *C*_t_ results derived from all experiments. These results underline the importance of the plastic tubes type continuity used during investigations. The *C*_t_ difference (type I–III) obtained for M2 (with *M. orale*) was 9.42, which is a significant quantity in the context of evaluation and sensitivity. 

#### 3.1.6. DAPI Test and Mycoplasma’s Culture on Agar 

The MG63 cells showed positive results in Test-1, but in Test-2 were positive only with primers specific for *M. arginini*. After a week, the microbiological culture established typical mycoplasma’s colonies as “fried-egg shaped colonies” that were observed under an inverted microscope at 100× and 400× magnification (Figure 4). Additionally, the DAPI test performed for the same MG63 sample was negative, and the examined cells did not show any phenotypic changes and problems with culture. 

#### 3.1.7. Quality Control Results of Cell Lines Obtained by Test-1 and Test-2 

The results of QC in term of *Mycoplasma*, *Acholeplasma*, and *Ureaplasma* sp contamination of mammal cells tested by Test-1 and -2 showed their practical application according to the aim of this study. Test-1 detected *Mycoplasma sp.* contamination in 27 DNA samples (33.33%) from nine different types of cell lines. Test-2 results confirmed all positive and negative results of Test-1. It showed contamination with M. *arginini* and *M. hyorhinis* in cells of ME18, MER, HeLa, KB-V1, L929, WISH, BJ, and A549. MG63 cells were contaminated only with *M. arginini.* No detected any DNA *M. orale, M. fermentans, M. genitalium, M. hominis, M. genitalium, M. salivarium, A. laidlawii,* and *U. urealyticum* were not detected in the remained cell lines with the use of Test-1 and Test-2 (Table 4). All tested samples (*n* = 27) of cell lines with GM13509, GM14467, HL60, h.FOB1.19, TK, MDBK, MCF, and GM21756 cells were negative but two A545 and WISH were positive (Figure 5). 

## 4. Discussion

Detection of cell culture contaminations caused by bacteria belonging to *Mycoplasma* sp., *Acholeplasma* sp., and *Ureaplasma* sp. that are common in human-environment requires careful detection and analysis, especially with using of qPCR methods [10]. To obtain reliable results only validated methods should be used. Several variants of qPCR techniques have been described so far for the investigation of cell culture contamination caused by *Mollicutes* [1,10,11,31]. 

Our study presents newly developed PCR assay to detect DNA of selected species from members of *Mycoplasma*, *Acholeplasma*, and *Ureaplasma* groups in the samples of cell cultures using a two-steps strategy with SYBR Green I. 

The first step provides screening of examined bacteria with Test-1 from members of *Mycoplasma*, *Acholeplasma*, and *Ureaplasma* groups in screening Test-1 by triplex qPCR (Test-1 v.1) or in three single qPCR (Test-1 v.2). The next step, Test-2, relies on species identification using primers choice based on the Test-1 results (Scheme 1). Application of Test-1 variants is optional and depends on the final goal. 

Proposed way of analysis reduces the time and costs of cell cultures microbial QC and facilitates testing because of applied commonly used fluorochrome type. 

The specificity of Test-1 has been shown initially in silico level by DNA sequence fragments identification that are characteristic only for selected species belonged to *Mollicutes*. The NCBI BLAST database and BioEdit software analyzes showed full sequence compatibility only for fragments where primers associate during annealing. The remaining parts of sequences were various and characteristic to each selected species (Figure 1a–c). The Beacon Designer software used for primer designing facilitated selecting the best primer pairs with the multi-parameter characteristics presented in Appendix A. A low level of ΔG values of hairpins and dimers decreased the formation of non-specific products during PCR. The in silico results initially performed in this study showed several amplicon sequences from the selected targets with sequence fragments specific only for the selected species to avoid cross reactivity. However, in Test-1 we used commonly chosen genes [3,10,32], including *16S–23S ITS*, *16S rRNA*, *23S RNA* genes (Appendix A) but in this study we designed three unique (Fw and Rv) M1, M2, M3 pair of primers for screening detection of 11 selected species belonged to the *Mycoplasma* sp., *Ureaplasma* sp., and *Acholeplasma* sp. with SYBR Green. In the case of specific amplicon detection in Test-1, characterized by Tm values demonstrated in Appendix A the next step, species identification, is possible using Test-2 with the use qPCR with species-specific primers selected from either M1, M2, or M3 group (Scheme 1a–c). Apart from species identification, Test-2 additionally confirmed Test-1 species selected from *Mycoplasma*, *Ureaplasma*, and *Acholeplasma* sp. Higher primer concentrations used in Test-1 in comparison to Test-2 were related with the higher annealing temperature needed to increase the specificity of Test-1. It was essential to ensure a sufficient sensitivity level of Test-1 as a screening test while simultaneously providing the highest possible specificity. There was no detected cross-reactivity with DNA of closely related of bacterial species and DNA derived from different mammal cultures (Figure 2 and Appendix A). Frequent quality control of cell cultures is essential for the reliability of results obtained in cell culture experiments performed in vitro. 

Analytical sensitivity of Test-1 amounted below 1 fg/mL (10^−9^ and 10^−10^ ng Table 1) of genomic DNA of tested species that means was slightly more sensitive than the other method described by Kazemiha et al. (with LOD about 3 fg/mL) [17]. The slight difference in the sensitivity of Test-1 was only seen for *A. laidlawii* and could be related to the formation of two specific qPCR fragments with different sizes (Table 1). Specificity and relative accuracy calculated for developed Test-1 and Test-2 showed full concordance with the commercial test used in this study as a comparison test for all tested gDNA.

According to pharmacopeial requirements in the range of *Mycoplasma/Ureaplasma/Acholeplasma* sp. for mammal cell cultures our new developed assay in time saving enabled microbial QC and species identification. 

The QC results obtained in this study concerned 17 types of tested cell lines (Figure 5) and showed contaminations caused by two mycoplasma’s species *M. arginini* and *M. hyorhinis* in eight of them. All positive in qPCR cell cultures presented discreet phenotypic changes and were undetectable by DAPI test. Therefore application other than unspecific tests for QC is advisable. Identification of *M. arginini* and *M. hyorhinis* in tested cells in our study may indicate for animal’s origin of detected contamination. In turn of the confirmation contamination caused by two the same bacterial species found in eight different types of cell lines may additionally indicate for the intra-laboratory spread. This two-steps assay seems to be more useful, than other without possibility of species identification. The knowledge of mycoplasma’s species or resistance pattern is required for the choice of optimal antibiotic treatment [33] especially in the case of infected cultures. However contaminated cell cultures should be discarded but special treatment can save of unique cell cultures. The knowledge about the *Mycoplasma* species can help in the choice of rationale antibiotic therapy. The advantage of screening tests is considerably lowering the costs and shortening the time of testing. The disadvantage of the screening methods is the lack of identification of bacterial species among contaminated cell cultures and related with that limit of information about of antibiotic choice for treatment.

Our study showed contamination of the old cell culture collection by two species, *M. arginini* and *M. hyorhinis*. Similarly, Kazemiha V.M. et al. in 2019 showed that 35% of tested cell lines were contaminated at least two *Mycoplasma* sp. (19%) and three (16%) [34]. They also showed differences under drug resistance among several *Mycoplasma* sp. 

Cell cultures contamination caused by typical microbials can be easily noticed by direct cloudiness observation in a growth medium or microscope. In such situation immediate removal of contaminated cultures and all environments disinfection should be performed. In contrast to typical microbial infections, *Mycoplasma*, *Acholeplasma*, and *Ureaplasma* sp. infections may be unnoticeable. This study shows an example of contamination evaluation difficulties with no cellular changes in MG63 cell cultures (Figure 4a,b). Discreet changes were visible upon microscopic evaluation only in MG63 cells after DAPI staining but only in extra visible light. Bacterial colonies found on the MABG agar with phenotypic *Mycoplasma* features allowed to verify inconsistent results obtained for this cell line in DAPI and qPCR. Test-1 with M1 and M2 primers was positive only for MG63 cells, but Test-2 identified *M. arginini* only in MG63 cells and no *Mycoplasma/Acholeplasma/Ureaplasma* sp. traces such as shown in L929 cells. We detected on agar with characteristic for *Mycoplasma* sp. growth (Figure 4i–k) and under an inverted microscope (Figure 4j–l) at 400× magnification. 

Summarizing, the choice of qPCR variant used during diagnostic process mainly depends on the aim of study, the type of biological specimen, type of fluorochrome used for amplicon detection, availability of special equipment, the spectrum of validated parameters, and the costs of examination [30]. Commercially available validated tests usually have limited application and are designed for the insufficient types of specimens or qPCR platforms, or have no efficient validation, allowing their use for a broad spectrum of applications [29]. The application of commercially available methods requires revalidation of some parameters, including accessible equipment, reagents with a fluorochrome and plastic tubes or other factors. 

## 5. Conclusions

A newly developed qPCR assay with SybrGreen I presented in this study composed with two tests (Test-1 and Test-2) can be used for mammalian cell cultures QC according to pharmacopeial requirements in order to detect contamination caused by selected *Mycoplasma*, *Acholeplasma* and *Ureaplasma* species, including *M. arginini*, *M. orale*, *M. hyorhinis*, *M. fermentans*, *M. genitalium*, *M. hominis*, *M. pneumoniae*, *M. salivarium*, *M. pirum*, *A. laidlawii*, and *U. urealyticum*. 

Based on our results Test-1, screening test allows to detect all specified species in one tube (Test-1 v.1) or in three tubes (Test-1 v.2) and identify mentioned microorganisms to the *Mycoplasma*, *Acholeplasma* or/and *Ureaplasma* three groups. The second, Test-2 is composed with single specific qPCR and to identify each mentioned above species. Application of Test-1 v.2 enables the reduction of single qPCR number to perform in Test-2. 

Both, Test-1 and Test-2 present comparable LOD level and can be used separately or in combination depending of the need. 

Based on our findings the proposed way of QC of mammalian cell cultures is competitive to the other qPCR assays performed based on of molecular probes by a lower costs and higher stability of reagents. 

## 6. Patents

Patent Application No. P.420423 in the Patent Office of the Republic of Poland.

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
