# Peer review of "Development of New PCR Assay with SYBR Green I for Detection of Mycoplasma, Acholeplasma, and Ureaplasma sp. in Cell Cultures"

_diagnostics, 2021, doi:10.3390/diagnostics11050876_

Round 1

Reviewer 1 Report

I reviewed this article in its previous version. I disagree in many aspects with the second reviewer who advised to reject the article. In particular, judging by one of their remarks, it creates the impression that it is not so critical to use in the work of one cell culture infected with any mycoplasma, since you can always do PCR with universal primers, determine the microorganism to a species as a result of sequencing, and then explain some twists and turns that can be caused by contamination. I strongly disagree with this position. In our experimental practice it has been clearly shown that mycoplasma infection can seriously affect, for example, signaling pathways associated with such factors as NF-kB, p53 or Nrf2. And this, in turn, can severely change intracellular processes in the eukaryotic cell. As for the approach using classical PCR with universal primers and subsequent sequencing of the amplificate and determining the bacteria to species before working with the cell culture - even if there is a sequencer in the organization, this should take longer and be somewhat more expensive than doubled PCR, as suggested the authors of the work. In addition, it is known that mycoplasmas respond differently to antibiotic therapy. In particular, if Mycoplasma hominis is not treated with roxithromycin, then Ureaplasma parvum / urealythicum is not always amenable to suppression with clindamycin or lincomycin. As for Acholeplasma, it can generally be eliminated, by and large, only with the help of tetracycline (and even then not always) or gentamicin, and tetracycline can have a strong negative effect on the cell cultures treated with it. Thus, I believe that it is better to first determine if there is contamination and what exactly. Then the culture should be treated with antibiotics, the choice of which should be calculated to cause minimal harm to the cells in the culture. Only after all these steps it is worth starting experimental work with culture. This is the aim the test system was created for. The authors have carefully approached the correction of the manuscript according to my previous comments, have gently responded to the second reviewer to their not always justified claims, so I continue to believe that the corrected and re-submitted article can be published in the journal.

Author Response

Dear Reviewer,

I would like to thank you very much for your evaluation of our manuscript, your strong support and a lot of examples in order to underline how important is the quality control in cell cultures especially in case of Mycoplasma contaminations. I have similar observations that these microorganisms are dangerous and it is really to difficult to get it rid of. Thank you so much also for interesting discussion concerning to the troubles with the choice of antibiotic to treatment infections caused by different species from Mycoplasma genera. I have underlined also additionally this aspect in the Discussion section of the manuscript.

Kind regards,

Jolanta Krzysztoń-Russjan

Reviewer 2 Report

Dear authors,

The examinations described in the manuscript „New way detection of Mycoplasma, Acholeplasma, and Ureaplasma sp. By quantitative PCR assay with SYBR Gree I” are interesting and important for cell line examination. Authors made many experiments to validate the newly developed method. The methods were used properly in the manuscript. But the results, discussion and conclusion must be improved.

Broad comments:

In the “Introduction” section methods are described incomprehensive and it is difficult to guess what message authors want to give.

  1. Methods used by the authors are described in very complicated way it should be simplified. Similarly the results are described in very complicated way. The authors should focus on describing the objectives of the study, without unnecessary comments. Some additional information should be put from the results to the discussion or from the discussion to the introduction. For example see the sentence in the lines 383-386.
  2. There are too many detailed information and the main objective of the study can be missed by the reader. The number of figures and tables and the amount of information in it is too large.
  3. The title of the manuscript should be changed. It can’t be used “New way detection”. I suggest to use for example: “Development of PCR assay with Sybr Green I for the detection of …”, it can be added the information - in what specimen this bacteria can be detected.
  4. Newly developed assay/test is not a new way. Authors should use more scientific language. They also shouldn’t name a new assay for the detection as a strategy.
  5. The subchapters “results”, “discussion” and “conclusions” are not written properly. In the results there are many aspects more proper for the discussion. The discussion is a description of methods and a conclusion is not clear and it contains some of the results once again. The conclusion should be rebuild completely.
  6. Authors should not use the abbreviation “MAU”, it should be used whole name of the bacteria genera.
  7. In the manuscript there is a lot of incomprehensible sentences, it is not known what authors want to tell.
  8. Authors describe in “Discussion” section “a new simply way”. The proposition of developed PCR is not simply, although it is easier than conventional culture it is complicated. Authors should consider this once again.
  9. Authors should remember that they examine the DNA of Mycoplasma, Acholeplasma, and Ureaplasma in the samples of cell lines not in the DNA template of cell line.

Specific comments:

Line 12: Delete “respectively”; change there and elsewhere “a new way” into for example “a developed qPCR”.

Line 14: Change “MAU” into “Mycoplasma, Acholeplasma, and Ureaplasma spp. DNA detection”.

Line 16: After “reduction of” add “using”.

Lines 17-19: The sentence is not clear – it is about microbial quality control testing? Rewrite.

Line 20 and elsewhere: Don’t use abbreviation “cn”.

Line 22 and elsewhere: Change “solving” into “saving”.

Lines 21-23: The sentence difficult to understand, rewrite. Especially “in species range” is not clear. Delete “DNA from” because more appropriate information is that there were tested cell cultures.

Lines 43-44: Rewrite the sentence.

Line 46: Wrong interpretation, reduced genome size is not a reason that this bacteria are difficult to detect under microscope - rewrite. Change “immediate” into “indirect”.

Line 56: Write “Mycoplasma” with italic.

Line 59: Explain why it is more valuable.

Line 60: Change “gene” into “DNA”.

Lines 61-62: The amplification is not a goal itself. It helps to detect the target DNA.

Lines 62-64: Why authors referred to this 2 tests. Explain the differences between it.

Line 65: Change “ PCR variants” into “molecular assays”.

Line 66-67: Not clear what authors want to tell. Rewrite.

Line 70: Change “An additional paper” into other/another research.

Lines 72-75: Authors mention different methods. Not clear what this sentence contribute to the introduction. Summarize.

Line 76: Add “s” to “sequence”.

Line 77: Change “genotyping through a gene fragment typing” into “genotyping of a gene fragment”.

Line 80: Change „detecting mycoplasmas” into “the detection of mycoplasmal”.

Lines 84-85: The microbial quality control can not be done for DNA. Rewrite.

Line 91 and elsewere: You can not write that you tested cryovials, authors tested 81 samples of seventeen different cell lines.

Line 93: Delete “with the number of tested cryovials” and “such”.

Line 122: Change “MAU gDNA” into “three gDNA”.

Line 130: Delete “ing” from “forwarding”.

Line 155: Delete “on different days”, it is obvious.

Lines 161-163: Correct the word :pairs”.

Line 171: Change “tests: into “examinations”.

Line 171, 187: Delete “on different dates”.

Lines 175-177: Add the information from where the samples were obtained.

Line 184: Change “MAU” into “isolated”.

Line 192: Delete “Human” from the beggining. Add “used as a template” after “GM1447”.

Line 193: Delete “we spiked”, add “was spiked” after “dilution”.

Line 194: Not clear what mean “serially cut”, rewrite.

Line 197: Compared with what?

Line 205: Add “manufacturers” after “according to the”.

Line 206: Add “incubated” before “at 95”.

Line 207: Lines 213-214: Not clear what was assessed, explain and rewrite.

Line 221, 229: Add description – is it the name of cell line?

Line 229: “cell culture ranges” explain what it mean.

Line 233: Add the information in what way the evaluation was done.

Line 243: Change “genetically” into “closely”.

Lines 244-246: Why this sentence was added to the results section? What it mean for the manuscript?

Line 248: Change into “Based on the DNA sequence analysis”.

Lines 252-253: Correct the word “primer”.

Line 255:Delete “_”.

Lines 280-284: It will be easier to read when the names of the gene will be put into brackets.

Scheme 1. The abbreviations are not explained. What such information as “FW, RV M1 M2 M3; KSMM” gives to the scheme – delete or explain. The scheme should be simplified. The clear information what is Test 1 and Test 2 should be added.

Line 327: word “fo” – correct.

Lines 331-332: Changed to “…without template DNA but with each…” and “ …resulted in no amplification of specific products”.

Lines 336-337: Remove this sentence to the discussion.

Line 432: Change “little bit” into “slightly”.

Line 355: The word “extremely” don’t math the context.

Line 365: Change :”new” into “qPCR”. Does authors describe a real reference method or is it used only for comparison?

Line 370: Change “included” into “used”.

Line 371-372: Does authors describe a real “gold standard” or is it used only for comparison? Change “new way” into “developed assay”. Change “ MAU subgroup …” into “Mycoplasma, Acholeplasma, and Ureaplasma spp. DNA detection and next, the species identification”.

Line 372-373: The sentence is incomprehensible, reader don’t known what authors want to tell.

Line 382: Change “specific changes” into “different conditions”.

Lines 383-386: Delete, not necessary for the “Results” section.

Line 387: It should be deleted “with the optimized parametres and same template amount”.

Line 397-399: Remove to the discussion.

Line 399-400: Delete.

Line 403: Change into “fried-egg shaped colonies”.

Line 405: Change “cryovial” into “sample”.

Line 408: Remove “tested” after “cells”.

Line 411: Change “causeb by” into “with”.

Line 412: Delete “but in”. Change into “MG63 cells were contaminated only with…”. Add “DNA” after “any”.

Line 413: Change into “U. urealyticum were not detected in the remained cell lines with the use of Test-1 and Test-2”.

Line 414: Change into “ All tested samples (n=27) of cell lines i.e. …”.

Line 415: Delete “next” and “only”.

Line 416-417: Replace to the “Discussion” section.

Figure 4: In my opinion it is not necessary, it is hard to read. Reader could have problem to read and understand it. The abbreviations are not explained. I suggest not to include it to the manuscript.

Figure 5: Explain what “MG63” and similar mean.

Line 472: Delete “captured”.

Figure 6: Change “cryovials” into “ samples of cell lines”.

Table 2: Correct the word “detection”.

Table 4: Change “contamination” into “presence” and “deposited in 30 cryovials” into “30 samples”.

Line 510: Change “species from” into “bacteria belonging to”.

Line 511: Sentence difficult to understand.

Line 514: Change from “species from MAU” into “members of Mycoplasma, Acholeplasma, and Ureaplasma groups”.

Line 515: “DNA templates” – authors should not write that they detect DNA of Mycoplasma, Acholeplasma, and Ureaplasma spp. in human and mammals DNA template but the DNA of Mycoplasma, Acholeplasma, and Ureaplasma spp. in the samples cell cultures.

Line 516: Beggin the sentence with “The first step provides screening of examined bacteria with Test-1…”.

Line 517: Change “allows for” into “relies on”.

Line 519: Add “of analysis” after “way”.

Line 531: Delete “most accurate MAU identification”, remove to the end “with Sybr Green”.

Line 534: Change “from the next” into “with the use”.

Line 539: Change “genetically most” into “DNA of closely”, delete “DNA standards”.

Line 540: What means “reasonable” in this context.

Line 542: Use abbrev. of femtogram.

Line 543: Change “MAU” into “tested species”.

Change “a little bit more” into more scientific language.

Line 544: Change “new” into “developed”.

Line 545: Change “gold standard” into “comparison”. Delete the name of test from the “Discussion”section.

Line 548: Change into: “…our new developed test/assay in time saving…”.

Line 550: These particular species are not seen in the Figure 6.

Lines 553-554: Rewrite, difficult to understand.

Line 556: “Screening tests” – add more information, what is the difference and disadvantage.

Lines 561-562: Difficult to understand. Authors mean differences or differentiation? Descrbe and rewrite.

Line 564: Change “of” into “under”, “that cases needs” into “In such situation”.

Line 569: Do not use “ultimately”.

Line 570: Delete “DNA from”. Change “confirmed this result with” into “identified”.

Line 572: Beggin the sentence with “We detected on agar with characteristic for Mycoplasma sp. growth”.

Line 576: After the “cost” add “of examination”.

Line 577: Add “s” to “type” and “platform”.

Line 574-580: Give references to support the thesis.

Line 580: What it mean? There is no summarise.

Author Response

Dear Reviewer,

Thank you so much for revision of the manuscript with ID: diagnostics-1210943. Responding to the posted remarks I would like to give you following answers (marked in blue color).

Broad comments:

In the “Introduction” section methods are described incomprehensive and it is difficult to guess what message authors want to give.

Answer: I have deleted a part of the Introduction concerned the typing methods (lines 71 - 81) as less related and I have corrected all mistakes that were placed in the list of the specific comments (placed below).

  1. Methods used by the authors are described in very complicated way it should be simplified. Similarly the results are described in very complicated way. The authors should focus on describing the objectives of the study, without unnecessary comments. Some additional information should be put from the results to the discussion or from the discussion to the introduction. For example see the sentence in the lines 383-386.

       Answer: I have deleted or replaced all sentences according to the list of details placed below.

  1. There are too many detailed information and the main objective of the study can be missed by the reader. The number of figures and tables and the amount of information in it is too large.

Answer: I have deleted Figure nr 4. I would like to explain that during development this new assay we have received a lot of information.

  1. The title of the manuscript should be changed. It can’t be used “New way detection”. I suggest to use for example: “Development of PCR assay with Sybr Green I for the detection of …”, it can be added the information - in what specimen this bacteria can be detected.

Answer: I have changed the title according to proposed suggestions. I would like to explain that presented way of detection is original because of the way construction included the spectrum of identified species and applied fluorochrome (SYBR Green I) that is very demanding to ensure the specificity of the amplicons

Newly developed assay/test is not a new way. Authors should use more scientific language. They also shouldn’t name a new assay for the detection as a strategy.

Answer: I have changed the sentence according to suggestions.

  1. The subchapters “results”, “discussion” and “conclusions” are not written properly. In the results there are many aspects more proper for the discussion. The discussion is a description of methods and a conclusion is not clear and it contains some of the results once again. The conclusion should be rebuild completely.

Answer: I have replaced or deleted some sentences according to Reviewer’s suggestions and some sentences that were duplicated. I have written a new Conclusions.

  1. Authors should not use the abbreviation “MAU”, it should be used whole name of the bacteria genera.

Answer: I have deleted MAU abbreviation (except in Scheme).

  1. In the manuscript there is a lot of incomprehensible sentences, it is not known what authors want to tell.

Answer: I have improved all of pointed sentences.

  1. Authors describe in “Discussion” section “a new simply way”. The proposition of developed PCR is not simply, although it is easier than conventional culture it is complicated. Authors should consider this once again.

Answer: I think that our proposition is easy to preform and with interpretation. It takes max 2 hours including DNA isolation.

  1. Authors should remember that they examine the DNA of Mycoplasma, Acholeplasma, and Ureaplasma in the samples of cell lines not in the DNA template of cell line.

Answer: Thank you for this notice, I have improved it in the manuscript.

Specific comments:

I have marked in yellow color all remarks on this list that I have improved in the manuscript, according to the suggestion. My answers I have marked in blue color.

Line 12: Delete “respectively”; change there and elsewhere “a new way” into for example “a developed qPCR”.

Line 14: Change “MAU” into “Mycoplasma, Acholeplasma, and Ureaplasma spp. DNA detection”.

Line 16: After “reduction of” add “using”.

Lines 17-19: The sentence is not clear – it is about microbial quality control testing? Rewrite.

Line 20 and elsewhere: Don’t use abbreviation “cn”.

Line 22 and elsewhere: Change “solving” into “saving”.

Lines 21-23: The sentence difficult to understand, rewrite. Especially “in species range” is not clear. Delete “DNA from” because more appropriate information is that there were tested cell cultures.

Lines 43-44: Rewrite the sentence.

Line 46: Wrong interpretation, reduced genome size is not a reason that this bacteria are difficult to detect under microscope - rewrite. Change “immediate” into “indirect”.

Line 56: Write “Mycoplasma” with italic.

Line 59: Explain why it is more valuable.

Line 60: Change “gene” into “DNA”.

Lines 61-62: The amplification is not a goal itself. It helps to detect the target DNA.

Lines 62-64: Why authors referred to this 2 tests. Explain the differences between it.

Line 65: Change “ PCR variants” into “molecular assays”.

Line 66-67: Not clear what authors want to tell. Rewrite.

Line 70: Change “An additional paper” into other/another research.

Lines 72-75: Authors mention different methods. Not clear what this sentence contribute to the introduction. Summarize.

Line 76: Add “s” to “sequence”.

Line 77: Change “genotyping through a gene fragment typing” into “genotyping of a gene fragment”.

Line 80: Change „detecting mycoplasmas” into “the detection of mycoplasmal”.

Lines 84-85: The microbial quality control can not be done for DNA. Rewrite.

Line 91 and elsewere: You can not write that you tested cryovials, authors tested 81 samples of seventeen different cell lines.

Line 93: Delete “with the number of tested cryovials” and “such”.

Line 122: Change “MAU gDNA” into “three gDNA”.

Line 130: Delete “ing” from “forwarding”.

Line 155: Delete “on different days”, it is obvious.

Lines 161-163: Correct the word :pairs”.

Line 171: Change “tests: into “examinations”.

Line 171, 187: Delete “on different dates”.

Lines 175-177: Add the information from where the samples were obtained.

Line 184: Change “MAU” into “isolated”.

Line 192: Delete “Human” from the beggining. Add “used as a template” after “GM1447”.

Line 193: Delete “we spiked”, add “was spiked” after “dilution”.

Line 194: Not clear what mean “serially cut”, rewrite.

Line 197: Compared with what?

Line 205: Add “manufacturers” after “according to the”.

Line 206: Add “incubated” before “at 95”.

Line 207: Lines 213-214: Not clear what was assessed, explain and rewrite.

Line 221, 229: Add description – is it the name of cell line?

Line 229: “cell culture ranges” explain what it mean.

Line 233: Add the information in what way the evaluation was done.

Line 243: Change “genetically” into “closely”.

Lines 244-246: Why this sentence was added to the results section? What it mean for the manuscript?

Line 248: Change into “Based on the DNA sequence analysis”.

Lines 252-253: Correct the word “primer”.

Line 255:Delete “_”.

Lines 280-284: It will be easier to read when the names of the gene will be put into brackets.

Scheme 1. The abbreviations are not explained. What such information as “FW, RV M1 M2 M3; KSMM” gives to the scheme – delete or explain. The scheme should be simplified. The clear information what is Test 1 and Test 2 should be added.

 Answer: The scheme was simplified and the abbreviations were explained.

Line 327: word “fo” – correct.

Lines 331-332: Changed to “…without template DNA but with each…” and “ …resulted in no amplification of specific products”.

Lines 336-337: Remove this sentence to the discussion.

Line 432: Change “little bit” into “slightly”.

Line 355: The word “extremely” don’t math the context.

Line 365: Change :”new” into “qPCR”. Does authors describe a real reference method or is it used only for comparison?

Answer: It is used only for comparison.

Line 370: Change “included” into “used”.

Line 371-372: Does authors describe a real “gold standard” or is it used only for comparison?

Answer: It is used only for comparison.

Change “new way” into “developed assay”. Change “ MAU subgroup …” into “Mycoplasma, Acholeplasma, and Ureaplasma spp. DNA detection and next, the species identification”.

Line 372-373: The sentence is incomprehensible, reader don’t known what authors want to tell.

Answer: I have deleted the last two sentences: “The proposed way is open and allow for MAU detection in different biological samples with good DNA quality. It is also helpful for monitoring of MAU detection in case of internal contamination to primary source identification” from this section.

Line 382: Change “specific changes” into “different conditions”.

Lines 383-386: Delete, not necessary for the “Results” section.

Line 387: It should be deleted “with the optimized parametres and same template amount”.

Line 397-399: Remove to the discussion.

Line 399-400: Delete.

Line 403: Change into “fried-egg shaped colonies”.

Line 405: Change “cryovial” into “sample”.

Line 408: Remove “tested” after “cells”.

Line 411: Change “causeb by” into “with”.

Line 412: Delete “but in”. Change into “MG63 cells were contaminated only with…”. Add “DNA” after “any”.

Line 413: Change into “U. urealyticum were not detected in the remained cell lines with the use of Test-1 and Test-2”.

Line 414: Change into “ All tested samples (n=27) of cell lines i.e. …”.

Line 415: Delete “next” and “only”.

Line 416-417: Replace to the “Discussion” section.

Figure 4: In my opinion it is not necessary, it is hard to read. Reader could have problem to read and understand it. The abbreviations are not explained. I suggest not to include it to the manuscript.

Answer: I have deleted Figure 4 from the manuscript.

Figure 5: Explain what “MG63” and similar mean.

Line 472: Delete “captured”.

Figure 6: Change “cryovials” into “ samples of cell lines”.

Table 2: Correct the word “detection”.

Table 4: Change “contamination” into “presence” and “deposited in 30 cryovials” into “30 samples”.

Line 510: Change “species from” into “bacteria belonging to”.

Line 511: Sentence difficult to understand.

Answer: I have improved the sentence,

Line 514: Change from “species from MAU” into “members of Mycoplasma, Acholeplasma, and Ureaplasma groups”.

Line 515: “DNA templates” – authors should not write that they detect DNA of Mycoplasma, Acholeplasma, and Ureaplasma spp. in human and mammals DNA template but the DNA of Mycoplasma, Acholeplasma, and Ureaplasma spp. in the samples cell cultures.

Line 516: Beggin the sentence with “The first step provides screening of examined bacteria with Test-1…”.

Line 517: Change “allows for” into “relies on”.

Line 519: Add “of analysis” after “way”.

Line 531: Delete “most accurate MAU identification”, remove to the end “with Sybr Green”.

Line 534: Change “from the next” into “with the use”.

Line 539: Change “genetically most” into “DNA of closely”, delete “DNA standards”.

Line 540: What means “reasonable” in this context.

Answer: I have changed this word into “frequent”

Line 542: Use abbrev. of femtogram.

Line 543: Change “MAU” into “tested species”.

Change “a little bit more” into more scientific language.

Line 544: Change “new” into “developed”.

Line 545: Change “gold standard” into “comparison”. Delete the name of test from the “Discussion”section.

Line 548: Change into: “…our new developed test/assay in time saving…”.

Line 550: These particular species are not seen in the Figure 6.

Answer: I have changed the place for Figure 6 (now Figure 5) in the sentence.

Lines 553-554: Rewrite, difficult to understand.

Answer: I have rewritten the sentence from: “Identified species suggest animal's origin source suspecting FBS or a growth medium supplement, as a potential initial source of contamination.” into: “Identification of M. arginini and M. hyorhinis in tested cells may indicate initially for animal’s origin of this contamination”.

Line 556: “Screening tests” – add more information, what is the difference and disadvantage.

Answer: I have added some information related to screening tests.

Lines 561-562: Difficult to understand. Authors mean differences or differentiation? Descrbe and rewrite.

Line 564: Change “of” into “under”, “that cases needs” into “In such situation”.

Line 569: Do not use “ultimately”.

Line 570: Delete “DNA from”. Change “confirmed this result with” into “identified”.

Line 572: Beggin the sentence with “We detected on agar with characteristic for Mycoplasma sp. growth”.

Line 576: After the “cost” add “of examination”.

Line 577: Add “s” to “type” and “platform”.

Line 574-580: Give references to support the thesis.

Line 580: What it mean? There is no summarise.

Answer: I have rewritten Conclusions part.

This manuscript is a resubmission of an earlier submission. The following is a list of the peer review reports and author responses from that submission.

Round 1

Reviewer 1 Report

I have a number of objections concerning the article. I don't see the point of the project. If any contamination occurs, it is almost exclusively monoculture, and then amplification with universal primers is sufficient. From the sequence of the PCR product, it is possible to unambiguously determine what kind of bacterium it is. This is important ex post when the results of the experiments are some twists that could be caused by contamination. Otherwise, the elimination of contamination from the strain is more important, and accurate identification does not help here. In addition, the work is written in a very amateur way and I cannot resist the impression that the authors do not understand the design and development of PCR diagnostics. This is evident from the repetition of sentences in the introduction, materials and methods, results, discussion and conclusions. They also provide details, for example on the properties of primers, which are not used. Other important data are clearly false like primers Tm. In addition, the primers are designed to conserve 16S RNA regions and therefore may not be specific enough. I can't imagine anyone using their work as a manual to identify Mycoplasma, Acholeplasma, and Ureaplasma sp.

Author Response

Rev.1

Thank you for the revision of the manuscript with ID: diagnostics-1086024. Responding to the posted objections I would like to give the following answers.

Rev. 1. “I don't see the point of the project.”

 Replay: The aim of the study is presented at the end of the Introduction section.

“Our study aimed to develop a sensitive and specific tool for easy detection and identification of Mycoplasma, Acholeplasma, and Ureaplasma selected species (MAU group) to perform microbial quality control of DNA derived from mammalian cell cultures. The selected range of identified species includes M. arginini, M. orale, M. hyorhinis, M. fermentans, M. genitalium, M. hominis, M. pneumoniae, M. salivarium, M. pirum, and two genetically related species, A. laidlawii, and U. urealyticum”. Additionally, a new way can be open for MAU group species investigations in biological samples derived from human or animal.

 Revi.1. “If any contamination occurs, it is almost exclusively monoculture, and then amplification with universal primers is sufficient.”  

Replay: In response to this comment, I have included the following explanation in the Discussion section. “Usually, applied tests for microbial quality detection of cell cultures control don’t identify the species among contaminated cell cultures because those are the screening tests. In infected cultures, it can usually be discarded; however, special treatment can save cells in unique cell cultures. In that case, the knowledge of Mycoplasma’s species or resistance pattern is required for optimal antibiotic therapy.

Our study showed contamination of the old cell culture collection by two species, M. arginini and M. hyorhinis. Similarly, Kazemiha V.M. et al. in 2019 showed that 35% of tested cell lines were contaminated at least two Mycoplasma sp. (19%) and three (16%). They also showed differentiation of drug resistance among different Mycoplasma sp.“

Rev. 1. From the sequence of the PCR product, it is possible to unambiguously determine what kind of bacterium it is. This is important ex post when the results of the experiments are some twists that could be caused by contamination.

Replay: Application of uncontaminated cell cultures by Mycoplasma, Acholeplasma, and Ureaplasma spp. (MAU) is better because it enables to obtain reliable results and saves time and costs. Performing quality control (QC) for all cell cultures before experiments and applying appropriate QC methods in routine work with cell line cultures is in line with Good Laboratory Practice. Our study presents an easy and reliable way for QC for MAU group detection that is only a part of QC that to be performed for cell line culture before using it for in vitro experiments. 

Rev. 1. In addition, the work is written in a very amateur way and I cannot resist the impression that the authors do not understand the design and development of PCR diagnostics. This is evident from the repetition of sentences in the introduction, materials and methods, results, discussion and conclusions.

Replay: The authors understand the design and development of PCR diagnostics. All redundant repetitions were removed.

Rev.1. They also provide details, for example on the properties of primers, which are not used.

Replay: The properties of primers are presented in Table 2a) and 2b). The data are explained below the Table 2a) to avoid repetition in the text.

Rev. 1. Other important data are clearly false like primers Tm. In addition, the primers are designed to conserve 16S RNA regions and therefore may not be specific enough.

Replay: Beacon Designer v. 7.91 software (Premier Biosoft, USA) calculated Tm primers' values and other properties of primers as noted below Table 2a, and 2b and concerned the values calculated only  by the software. The experimentally designated Tm values were presented the Table 3a, and 3b respectively. There were observed differences between Tm values obtained by software and after validation in experiments. Due to this ambiguity, I have rejected Tm values obtained in silico from Tables 2a and 2b and I have transferred Tables 2a, 2b, and 3a 3b to the Supplementary file to reduce presented details.

Rev. 1. In addition, the primers are designed to conserve 16S RNA regions and therefore may not be specific enough.

Replay: To answer this objection, I would like to explain that presented new way of Mycoplasma, Acholeplasma, and Ureaplasma sp. detection was validated to identify selected (MAU) species. Additionally, to increase the specificity of these primers temperature of annealing of those primes was an increase to 65áµ’C. Test-1, Test-2 verify the results and identify the species of bacteria from the MAU group in case of positive results. Especially considering that primers from Test-1 might also detect some plant, arthropods, or other animal groups Mycoplasmas in case of positive Test-1 and negative Test-2, the resulting amplicon should be sequenced and identified by sequence alignment in BLAST.

Reviewer 2 Report

The manuscript represents a meticulous work done. Mycoplasmas can cause not only severe diseases in weak organisms, but also they are one of the main problems in cell cultivation. Thus, quick and sensitive detection of these contaminants may become a useful instrument for cell biologists all over the world. This is what I must note - the authors applied for a patent (No. P.420423) in the Patent Office of the Republic of Poland, so this work will have additional approvement. Overall, the data is worth to be published after moderate revision. Here I present some inaccurances which I've found through the text, and some questions to the authors to answer.

1) Please, re-check the manuscript carefully for mistakes or misprints. To my opinion, they are present in the paper in some quantity. For example, these are on lines 29 (lack of comma), 68 (singe or SINGLE?), 75 (particulars or PARTICULAR?), 84 (culture or CULTIVATION?), 205-206 (double "sterile"), 213 (extra closing parenthesis), 247 (OF study), 289 (double parenthesis), 297 (unnecessary dot), 333 (It were calculated), 346 (M, M2, and M3 = M1?), 347 (anazlyzed), 362 (of a positive results), 396 (type = types), 481 (lack of semicolon), 489-495 (lack of closing parentheses), 507 (analysis OF results), (bsense = Absense in the Scheme 1), Table 8 - line 53 (dots instead of commas), etc. It's a technical problem, so this also can be done during correction procedures before publication. Moderate English revision is also required, especially for the Discussion chapter (This results, primer of pairs, be8low, and so on).

2) Please decipher cn/ml (for example, "copy numbers per ml") in the Abstract.

3) Could you please show that data (mentioned in line 140), for example, in the Supplement file?

4) Chapter 3.1.5 and Table 8. Is it possible to show here the names of plastic tube manufacturers? It is a really interesting information for researchers. 

5) Using of dark blue color in Figure 1 is debatable. I think it would be better to use something more contrasting with black.

6) The legends on Figures 2-5, 7-9 are too small; it is difficult to see anything there; please, enlarge them in size.

7) Figure captions (Fig.2 and 4) are overloaded with information; you can reduce them painlessly by deleting repeats.

8) I can't see (a) or (c) on Figure 6. And why does it not contain a (b) part? Maybe the caption does not concern to this Figure? Moreover, the low part of the Figure should be changed in size anyway.

9) Table 3a, lines 7-9 in the caption. Why are these Tm values not in the Table? And why are other data not presented for these species according to the whole table?

In result, I think the paper is overwhelmed with Figures and Tables. I would recommend to transfer some of them to the Supplement file. Which Figures to choose is at the discretion of the authors. I would recommend to remove from the main manuscript at least Tables 1, 2a, and 2b. Additionally, I would recommend to distribute Figures, Tables and the Scheme through the text to places where they will be appropriate. It will be easer to perceive the information then, instead of concentration of them in one chapter, to my opinion.

Author Response

Rev.2

Thank you for the revision of the manuscript with ID: diagnostics-1086024. Responding to the posted objections I would like to give the following answers.

Rev. 2.

  • Please, re-check the manuscript carefully for mistakes or misprints. To my opinion, they are present in the paper in some quantity. For example, these are on lines 29 (lack of comma), 68 (singe or SINGLE?), 75 (particulars or PARTICULAR?), 84 (culture or CULTIVATION?), 205-206 (double "sterile"), 213 (extra closing parenthesis), 247 (OF study), 289 (double parenthesis), 297 (unnecessary dot), 333 (It were calculated), 346 (M, M2, and M3 = M1?), 347 (anazlyzed), 362 (of a positive results), 396 (type = types), 481 (lack of semicolon), 489-495 (lack of closing parentheses), 507 (analysis OF results), (bsense = Absense in the Scheme 1), Table 8 - line 53 (dots instead of commas), etc. It's a technical problem, so this also can be done during correction procedures before publication. Moderate English revision is also required, especially for the Discussion chapter (This results, primer of pairs, be8low, and so on).

Replay: Thank you for noticing all these mistakes. I have corrected all of these.

  • Please decipher cn/ml (for example, "copy numbers per ml") in the Abstract.

Replay: I have corrected this abbreviation in the abstract.

  • Could you please show that data (mentioned in line 140), for example, in the Supplement file?

Replay: There is showed M1, M2, and M3 primer optimization in the Supplement file.

  • Chapter 3.1.5 and Table 8. Is it possible to show here the names of plastic tube manufacturers? It is a really interesting information for researchers. 

Replay: I have complete information about tube types and manufacturers below Table 8.

  • Using of dark blue color in Figure 1 is debatable. I think it would be better to use

something more contrasting with black.

Replay: I have a technical problem with a change of color.

  • The legends on Figures 2-5, 7-9 are too small; it is difficult to see anything there; please, enlarge them in size.

Replay: I have enlarged the Figures mention above. 

7) Figure captions (Fig.2 and 4) are overloaded with information; you can reduce them painlessly by deleting repeats.

Replay: I have reduced description under Figures 2 and -4.

  • I can't see (a) or (c) on Figure 6. And why does it not contain a (b) part? Maybe the caption does not concern to this Figure? Moreover, the low part of the Figure should be changed in size anyway.

Replay: I have changed the size of the consolidated chart and also changed the description to the appropriate.

9) Table 3a, lines 7-9 in the caption. Why are these Tm values not in the Table? And why are other data not presented for these species according to the whole table?

Replay: All Tm values are presented in Table 3a. The other data are not shown for these species because DNA dilution curves were not performed for these bacterial species.

The Tm values ​​reported are the mean values ​​obtained from at least five independent replicates of the quality control studies using the listed DNA standards.

Rev. 2: In result, I think the paper is overwhelmed with Figures and Tables. I would recommend to transfer some of them to the Supplement file. Which Figures to choose is at the discretion of the authors. I would recommend to remove from the main manuscript at least Tables 1, 2a, and 2b. Additionally, I would recommend to distribute Figures, Tables and the Scheme through the text to places where they will be appropriate. It will be easier to perceive the information then, instead of concentration of them in one chapter, to my opinion.

 Replay: I have moved Tables nr 1, 2a, 2b, 4, 5, 8, and Figures nr 4, 5, 7 into the Supplement file

Reviewer 3 Report

The manuscript entitled “New way detection of Mycoplasma, Acholeplasma, and Ureaplasma sp. by qPCR assay with SYBR Green I” by Jolanta et al. provides a detailed description of a proof-of-concept two step quantitative PCR (qPCR)-based method (Test-1 and Test-2) for the detection and quantification (Test-1 ) and species identification (Test-2) of DNA from a variety of bacteria, such as Mycoplasma, Acholeplasma, and Ureaplasma sp., (including subgroups). The main part of the study is dedicated to the set-up of the Test-1 which employ wide range primers, contributing to the definition of the best analytical parameters. Then, Test-2 has also been developed in order to specifically identify bacterial subgroups employing species-specific primers. A validation of the qPCR protocols in bacterial positive cell lines has also been performed.

The authors used a comparison of methods strategy with PromoKine qPCR Test Kit I/RT. Developmental criteria were, among others, limit of detection and quantification, efficiency, linearity as well as precision, robustness, sensitivity, specificity and accuracy. The authors conclude that the evaluation of Test-1 and Test-2 performances and comparability of results were promising. However, in order to demonstrate the clinical application of the assay, I encourage the validation of the method by reproducibility and validation on a large number of clinical samples.

Although the methods section can be shortened, the manuscript is in general well written and the results are well discussed. With exceptions, figures are demonstrative. In my opinion, the manuscript can be accepted after minor revision. I have suggested several improvements.

Major concerns

  1. In order to better emphasize the reliability of the novel method described in this manuscript, a comparison with the standard methods reported before for the detection of Mycoplasma, Acholeplasma, and Ureaplasma DNAs should be included in the discussion section.
  2. I suggest to reduce the methods section and the number of tables while only maintaining fundamental sections. Additional parts can be included as supplemental file. Several parts are difficult to read. For instance the primer design/optimization section as well as table 1 (which I think is secondary for the purpose of the manuscript) can be moved to supplemental.
  3. Although novel and sensitive, the method is based on a two consecutive qPCR runs with numerous primer pairs, so I suggest to mitigate the “rapid detection and identification” points. Alternatively, the “time solving” meaning should be better discussed.
  4. Limitations/strengths of the novel protocol should be included

Specific comments

“Quantitative PCR” instead of “qPCR” could sound better in the title.

Line 14 “qPCR (quantitative polymerase chain reaction)”, and lines 68-69 --> in general, acronyms or abbreviations should be reported in parenthesis following the phrase. For instance: quantitative polymerase chain reaction (qPCR). Please revise the text accordingly.

Line 20 cn/ml should be described when quoted the first time. Copy number/milliliter?

Lines 66-77---> A quantitative PCR-based method which employ primers detecting DNA sequences from Ureaplasma parvum and urealiticum (GAP gene) and Mycoplasma genitalium and hominis (ureC gene) has recently been reported in a study conducted on clinical samples, including tissue (chorionic villi) and blood samples from pregnant females (PMID: 30078192). An additional paper describing a highly sensitive multiplexed qPCR for the detection of DNA belonging to Mycoplasma pneumoniae and other bacteria has also been reported (PMID: 20151681). For completeness of information, these studies should be quoted in the introduction section.

Page 2 of 33, Line 106 “amounted be8low 1 femtogram” ---> below?

Figures

Fig 4, 5, panels should be enlarged.

Fig 7, titles should be enlarged.

Author Response

Rev. 3

….Although the methods section can be shortened, the manuscript is in general well written and the results are well discussed. With exceptions, figures are demonstrative. In my opinion, the manuscript can be accepted after minor revision. I have suggested several improvements.

Rev. 3 suggestion. I encourage the validation of the method by reproducibility and validation on a large number of clinical samples.

Replay. I have just finished the next study (manuscript in preparation) related to in vitro investigation of M. arginini influence on energy metabolism and cellular stress of lymphoid cells with HTT gene mutation derived from Huntington’s diseases (HD) subjects in comparison to control cells without mutation. Next part of study will concern investigation of Mycoplasma, Ureaplasma and Acholeplasma traces in blood samples derived from subject with Huntington’s diseases and subjects with other neurodegenerative diseases.

Rev. 3. Major concerns:

 In order to better emphasize the reliability of the novel method described in this manuscript, a comparison with the standard methods reported before for the detection of Mycoplasma, Acholeplasma, and Ureaplasma DNAs should be included in the discussion section.

Replay:

I have included a comparison between the standard method and the novel method suggested in the discussion section.

  1. I suggest to reduce the methods section and the number of tables while only maintaining fundamental sections. Additional parts can be included as supplemental file. Several parts are difficult to read. For instance the primer design/optimization section as well as table 1 (which I think is secondary for the purpose of the manuscript) can be moved to supplemental.

Replay: I have reduced the method section and table nr  ,,,,,,,,,,,    moved to Supplemental sections

  1. Although novel and sensitive, the method is based on a two consecutive qPCR runs with numerous primer pairs, so I suggest to mitigate the “rapid detection and identification” po ints. Alternatively, the “time solving” meaning should be better discussed.

Replay: I have corrected the indicated term.

 Limitations/strengths of the novel protocol should be included

Replay: This aspect was more underlined in the Discussion section.

“Quantitative PCR” instead of “qPCR” could sound better in the title.

 Replay: I applied a change to the title as suggested above.

Line 14 “qPCR (quantitative polymerase chain reaction)”, and lines 68-69 --> in general, acronyms or abbreviations should be reported in parenthesis following the phrase. For instance: quantitative polymerase chain reaction (qPCR). Please revise the text accordingly.

 Line 20 cn/ml should be described when quoted the first time. Copy number/milliliter?

 Lines 66-77---> A quantitative PCR-based method which employ primers detecting DNA sequences from Ureaplasma parvum and urealiticum (GAP gene) and Mycoplasma genitalium and hominis (ureC gene) has recently been reported in a study conducted on clinical samples, including tissue (chorionic villi) and blood samples from pregnant females (PMID: 30078192). An additional paper describing a highly sensitive multiplexed qPCR for the detection of DNA belonging to Mycoplasma pneumoniae and other bacteria has also been reported (PMID: 20151681). For completeness of information, these studies should be quoted in the introduction section.

Replay: All these remarks were completed in indicated places in the manuscript.

Page 2 of 33, Line 106 “amounted be8low 1 femtogram” ---> below?

Replay: All misprints I have corrected

Figures

Fig 4, 5, panels should be enlarged.

Fig 7, titles should be enlarged.

Replay: All these figures were enlarged. Additionally, all were renumerated because two of them I have moved to the Supplementary file.

Round 2

Reviewer 1 Report

I still do not see the purpose of this project, even though it is written at the….

„Our study aimed to develop a sensitive and specific tool for easy detection and identification of Mycoplasma, Acholeplasma, and Ureaplasma selected species (MAU group) to perform microbial quality control of DNA derived from mammalian cell cultures“.

Any contamination can be detected by amplification with universal primers and sequencing. From the sequence of the PCR product, it is possible to unambiguously determine what kind of bacterium it is

I do not believe that this article can be publish in any serious journal.